# Estimating Predictive Rate–Distortion Curves via Neural Variational Inference

**DOI:** 10.3390/e21070640

**Published:** 2019-06-28

**Authors:** Michael Hahn, Richard Futrell

**Affiliations:** 1Department of Linguistics, Stanford University, Stanford, CA 94305, USA; 2Department of Language Science, University of California, Irvine, CA 92697, USA

**Keywords:** Predictive Rate–Distortion, natural language, information bottleneck, neural variational inference

## Abstract

The Predictive Rate–Distortion curve quantifies the trade-off between compressing information about the past of a stochastic process and predicting its future accurately. Existing estimation methods for this curve work by clustering finite sequences of observations or by utilizing analytically known causal states. Neither type of approach scales to processes such as natural languages, which have large alphabets and long dependencies, and where the causal states are not known analytically. We describe Neural Predictive Rate–Distortion (NPRD), an estimation method that scales to such processes, leveraging the universal approximation capabilities of neural networks. Taking only time series data as input, the method computes a variational bound on the Predictive Rate–Distortion curve. We validate the method on processes where Predictive Rate–Distortion is analytically known. As an application, we provide bounds on the Predictive Rate–Distortion of natural language, improving on bounds provided by clustering sequences. Based on the results, we argue that the Predictive Rate–Distortion curve is more useful than the usual notion of statistical complexity for characterizing highly complex processes such as natural language.

## 1. Introduction

Predicting the future from past observations is a key problem across science. Constructing models that predict future observations is a fundamental method of making and testing scientific discoveries. Understanding and predicting dynamics has been a fundamental goal of physics for centuries. In engineering, devices often have to predict the future of their environment to perform efficiently. Living organisms need to make inferences about their environment and its future for survival.

Real-world systems often generate complex dynamics that can be hard to compute. A biological organism typically will not have the computational capacity to perfectly represent the environment. In science, measurements have limited precision, limiting the precision with which one can hope to make predictions. In these settings, the main goal will typically be to get good prediction at low computational cost.

This motivates the study of models that try to extract those key features of past observations that are most relevant to predicting the future. A general information-theoretic framework for this problem is provided by Predictive Rate–Distortion [1,2], also known as the past-future information bottleneck [3]. The Predictive Rate–Distortion trade-off seeks to find an encoding of past observations that is maximally informative about future observations, while satisfying a bound on the amount of information that has to be stored. More formally, this framework trades off prediction loss in the future, formalized as cross-entropy, with the cost of representing the past, formalized as the mutual information between the past observations and the compressed representations of the past. Due to its information-theoretic nature, this framework is extremely general and applies to processes across vastly different domains. It has been applied to linear dynamical systems [3,4], but is equally meaningful for discrete dynamical systems [2]. For biological systems that make predictions about their environment, this corresponds to placing an information-theoretic constraint on the computational cost used for conditioning actions on past observations [5].

The problem of determining encodings that optimize the Predictive Rate–Distortion trade-off has been solved for certain specific kinds of dynamics, namely for linear dynamic systems [3] and for processes whose predictive dynamic is represented exactly by a known, finite Hidden Markov Model [2]. However, real-world processes are often more complicated. When the dynamics are known, a representing Hidden Markov Model may still be extremely large or even infinite, making general-purpose automated computation difficult. Even more importantly, the underlying dynamics are often not known exactly. An organism typically does not have access to the exact dynamics of its surroundings. Similarly, the exact distribution of sentences in a natural language is not known exactly, precluding the application of methods that require an exact model. Such processes are typically only available implicitly, through a finite sample of trajectories.

Optimal Causal Filtering (OCF, Still et al. [6]) addresses the problem of estimating Predictive Rate–Distortion from a finite sample of observation trajectories. It does so by constructing a matrix of observed frequencies of different pairs of past and future observations. However, such a method faces a series of challenges [2]. One is the curse of dimensionality: Modeling long dependencies requires storing an exponential number of observations, which quickly becomes intractable for current computation methods. This exponential growth is particularly problematic when dealing with processes with a large state space. For instance, the number of distinct words in a human language as found in large-scale text data easily exceeds 1×$10^{5}$, making storing and manipulating counts of longer word sequences very challenging. A second challenge is that of overfitting: When deploying a predictive model constructed via OCF on new data to predict upcoming observations, such a model can only succeed when the past sequences occurred in the sample to which OCF was applied. This is because OCF relies on counts of full past and future observation sequences; it does not generalize to unseen past sequences.

Extrapolating to unseen past sequences is possible in traditional time series models representing processes that take continuous values; however, such methods are less easily applied to discrete sequences such as natural language. Recent research has seen a flurry of interest in using flexible nonlinear function approximators, and in particular recurrent neural networks, which can handle sequences with discrete observations. Such machine learning methods provide generic models of sequence data. They are the basis of the state of the art by a clear and significant margin for prediction in natural language [7,8,9,10]. They also have been successfully applied to modeling many other kinds of time series found across disciplines [11,12,13,14,15,16,17,18].

We propose Neural Predictive Rate–Distortion (NPRD) to estimate Predictive Rate–Distortion when only a finite set of sample trajectories is given. We use neural networks both to encode past observations into a summary code, and to predict future observations from it. The universal function approximation capabilities of neural networks enable such networks to capture complex dynamics, with computational cost scaling only linearly with the length of observed trajectories, compared to the exponential cost of OCF. When deploying on new data, such a neural model can generalize seamlessly to unseen sequences, and generate appropriate novel summary encodings on the fly. Recent advances in neural variational inference [19,20] allow us to construct predictive models that provide almost optimal predictive performance at a given rate, and to estimate the Predictive Rate–Distortion trade-off from such networks. Our method can be applied to sample trajectories in an off-the-shelf manner, without prior knowledge of the underlying process.

In Section 2, we formally introduce Predictive Rate–Distortion, and discuss related notions of predictive complexity. In Section 3, we describe the prior method of Optimal Causal Filtering (OCF). In Section 4, we describe our method NPRD. In Section 5, we validate NPRD on processes whose Predictive Rate–Distortion is analytically known, showing that it finds essentially optimal predictive models. In Section 6, we apply NPRD to data from five natural languages, providing the first estimates of Predictive Rate–Distortion of natural language.

## 2. Predictive Rate–Distortion

We consider stationary discrete-time stochastic processes ${\left(X_{t}\right)}_{t\in Z}$, taking values in a state space *S*. Given a reference point in time, say T=0, we are interested in the problem of predicting the future of $X_{t\geq 0}=\left(X_{0},X_{1},X_{2},\ldots \right)$ from the past $X_{t<0}=\left(\ldots ,X_{-2},X_{-1}\right)$. In general—unless the observations $X_{t}$ are independent—predicting the future of the process accurately will require taking into account the past observations. There is a trade-off between the accuracy of prediction, and how much information about the past is being taken into account. On one extreme, not taking the past into account at all, one will not be able to take advantage of the dependencies between past and future observations. On the other extreme, considering the entirety of the past observations $X_{t\leq 0}$ can require storing large and potentially unbounded amounts of information. This trade-off between information storage and prediction accuracy is referred to as Predictive Rate–Distortion (PRD) [2]. The term rate refers to the amount of past information being taken into account, while distortion refers to the degradation in prediction compared to optimal prediction from the full past.

The problem of Predictive Rate–Distortion has been formalized by a range of studies. A principled and general formalization is provided by applying the Information Bottleneck idea [2,6,21]: We will write $\overset{\leftarrow }{X}$ for the past $X_{<0}$, and $\vec{X}$ for the future $X_{\geq 0}$, following [2]. We consider random variables *Z*, called codes, that summarize the past and are used by the observer to predict the future. Formally, *Z* needs to be independent from the future $\vec{X}$ conditional on the past $\overset{\leftarrow }{X}$: in other words, *Z* does not provide any information about the future except what is contained in the past. Symbolically: (1)$$Z⊥\vec{X}|\overset{\leftarrow }{X}.$$

This is equivalent to the requirement that $Z↔\overset{\leftarrow }{X}↔\vec{X}$ be a Markov chain. This formalizes the idea that the code is computed by an observer from the past, without having access to the future. Predictions are then made based only on *Z*, without additional access to the past $\overset{\leftarrow }{X}$.

The rate of the code *Z* is the mutual information between *Z* and the past: $I[Z,\overset{\leftarrow }{X}]$. By the Channel Coding Theorem, this describes the channel capacity that the observer requires in order to transform past observations into the code *Z*.

The distortion is the loss in predictive accuracy when predicting from *Z*, relative to optimal prediction from the full past $\overset{\leftarrow }{X}$. In the Information Bottleneck formalization, this is equivalent to the amount of mutual information between past and future that is *not* captured by *Z* [22]: (2)$$I[\overset{\leftarrow }{X},\vec{X}|Z].$$

Due to the Markov condition, the distortion measure satisfies the relation
(3)$$I\left[\overset{\leftarrow }{X},\vec{X}|Z\right]=I\left[\overset{\leftarrow }{X},\vec{X}\right]-I\left[Z,\vec{X}\right],$$
i.e., it captures how much less information *Z* carries about the future $\vec{X}$ compared to the full past $\overset{\leftarrow }{X}$. For a fixed process ${\left(X_{t}\right)}_{t}$, choosing *Z* to minimize the distortion is equivalent to maximizing the mutual information between the code and the future: (4)$$I[Z,\vec{X}].$$

We will refer to (Equation 4) as the predictiveness of the code *Z*.

The rate–distortion trade-off then chooses *Z* to minimize distortion at bounded rate: (5)$$\min_{Z:I[\overset{\leftarrow }{X},Z]\leq d}I\left[\overset{\leftarrow }{X},\vec{X}|Z\right]$$
or—equivalently—maximize predictiveness at bounded rate: (6)$$\max_{Z:I[\overset{\leftarrow }{X},Z]\leq d}I\left[Z,\vec{X}\right].$$

Equivalently, for each λ≥0, we study the problem
(7)$$\max_{Z}\left(I\left[Z,\vec{X}\right]-\lambda \cdot I\left[\overset{\leftarrow }{X},Z\right]\right),$$
where the scope of the maximization is the class of all random variables *Z* such that $Z\to \overset{\leftarrow }{X}\to \vec{X}$ is a Markov chain.

The objective (Equation 7) is equivalent to the Information Bottleneck [21], applied to the past and future of a stochastic process. The coefficient λ indicates how strongly a high rate $I[\overset{\leftarrow }{X},Z]$ is penalized; higher values of λ result in lower rates and thus lower values of predictiveness.

The largest achievable predictiveness $I[Z,\vec{X}]$ is equal to $I[\overset{\leftarrow }{X},\vec{X}]$, which is known as the excess entropy of the process [23]. Due to the Markov condition (Equation 1) and the Data Processing Inequality, predictiveness of a code *Z* is always upper-bounded by the rate: (8)$$I\left[Z,\vec{X}\right]\leq I\left[\overset{\leftarrow }{X},Z\right].$$

As a consequence, when λ≥1, then (Equation 7) is always optimized by a trivial *Z* with zero rate and zero predictiveness. When λ=0, any lossless code optimizes the problem. Therefore, we will be concerned with the situation where λ∈(0,1).

### 2.1. Relation to Statistical Complexity

Predictive Rate–Distortion is closely related to Statistical Complexity and the ϵ-machine [24,25]. Given a stationary process $X_{t}$, its causal states are the equivalence classes of semi-infinite pasts $\overset{\leftarrow }{X}$ that induce the same conditional probability over semi-infinite futures $\vec{X}$: Two pasts $\overset{\leftarrow }{X},{\overset{\leftarrow }{X}}^{\prime }$ belong to the same causal state if and only if $P\left(x_{1\ldots k}|\overset{\leftarrow }{X}\right)=P\left(x_{1\ldots k}|{\overset{\leftarrow }{X}}^{\prime }\right)$ holds for all finite sequences $x_{0\ldots k}$ (k∈N). Note that this definition is not measure-theoretically rigorous; such a treatment is provided by Löhr [26].

The causal states constitute the state set of a a Hidden Markov Model (HMM) for the process, referred to as the ϵ-machine [24]. The statistical complexity of a process is the state entropy of the ϵ-machine. Statistical complexity can be computed easily if the ϵ-machine is analytically known, but estimating statistical complexity empirically from time series data are very challenging and seems to at least require additional assumptions about the process [27].

Marzen and Crutchfield [2] show that Predictive Rate–Distortion can be computed when the ϵ-machine is analytically known, by proving that it is equivalent to the problem of compressing causal states, i.e., equivalence classes of pasts, to predict causal states of the backwards process, i.e., equivalence classes of futures. Furthermore, [6] show that, in the limit of λ→0, the code *Z* that optimizes Predictive Rate–Distortion (Equation 7) turns into the causal states.

### 2.2. Related Work

There are related models that represent past observations by extracting those features that are relevant for prediction. Predictive State Representations [28,29] and Observable Operator Models [30] encode past observations as sets of predictions about future observations. Rubin et al. [31] study agents that trade the cost of representing information about the environment against the reward they can obtain by choosing actions based on the representation. Relatedly, Still [1] introduces a Recursive Past Future Information Bottleneck where past information is compressed repeatedly, not just at one reference point in time.

As discussed in Section 2.1, estimating Predictive Rate–Distortion is related to the problem of estimating statistical complexity. Clarke et al. [27] and Still et al. [6] consider the problem of estimating statistical complexity from finite data. While statistical complexity is hard to identify from finite data in general, Clarke et al. [27] introduces certain restrictions on the underlying process that make this more feasible.

## 3. Prior Work: Optimal Causal Filtering

The main prior method for estimating Predictive Rate–Distortion from data are Optimal Causal Filtering (OCF, Still et al. [6]). This method approximates Predictive Rate–Distortion using two approximations: first, it replaces semi-infinite pasts and futures with bounded-length contexts, i.e., pairs of finite past contexts $\left(\overset{M\leftarrow }{X}:=X_{-M}\ldots X_{-1}\right)$ and future contexts $\left(\overset{\to M}{X}:=X_{0}\ldots X_{M-1}\right)$ of some finite length *M*.(It is not crucial that past and future contexts have the same lengths, and indeed Still et al. [6] do not assume this). (We do assume equal length throughout this paper for simplicity of our experiments, though nothing depends on this). The PRD objective (Equation 7) then becomes (Equation 9), aiming to predict length-*M* finite futures from summary codes *Z* of length-*M* finite pasts: (9)$$\max_{Z:Z⊥\overset{\to M}{X}|\overset{M\leftarrow }{X}}\left(I\left[Z,\overset{\to M}{X}\right]-\lambda \cdot I\left[\overset{M\leftarrow }{X},Z\right]\right).$$

Second, OCF estimates information measures directly from the observed counts of $\left(\overset{M\leftarrow }{X}\right),\left(\overset{\to M}{X}\right)$ using the plug-in estimator of mutual information. With such an estimator, the problem in (Equation 9) can be solved using a variant of the Blahut–Arimoto algorithm [21], obtaining an encoder $P(Z|\overset{M\leftarrow }{X})$ that maps each observed past sequence $\overset{M\leftarrow }{X}$ to a distribution over a (finite) set of code words *Z*.

Two main challenges have been noted in prior work: first, solving the problem for a finite empirical sample leads to overfitting, overestimating the amount of structure in the process. Still et al. [6] address this by subtracting an asymptotic correction term that becomes valid in the limit of large *M* and λ→0, when the codebook $P(Z|\overset{\leftarrow }{X})$ becomes deterministic, and which allows them to select a deterministic codebook of an appropriate complexity. This leaves open how to obtain estimates outside of this regime, when the codebook can be far from deterministic.

The second challenge is that OCF requires the construction of a matrix whose rows and columns are indexed by the observed past and future sequences [2]. Depending on the topological entropy of the process, the number of such sequences can grow as ${\left|A\right|}^{M}$, where *A* is the set of observed symbols, and processes of interest often do show this exponential growth [2]. Drastically, in the case of natural language, *A* contains thousands of words.

A further challenge is that OCF is infeasible if the number of required codewords is too large, again because it requires constructing a matrix whose rows and columns are indexed by the codewords and observed sequences. Given that storing and manipulating matrices greater than $10^{5}\times 10^{5}$ is currently not feasible, a setting where $I\left[\overset{\leftarrow }{X},Z\right]>\log10^{5}\approx 11.5$ cannot be captured with OCF.

## 4. Neural Estimation via Variational Upper Bound

We now introduce our method, Neural Predictive Rate–Distortion (NPRD), to address the limitations of OCF, by using parametric function approximation: whereas OCF constructs a codebook mapping between observed sequences and codes, we use general-purpose function approximation estimation methods to compute the representation *Z* from the past and to estimate a distribution over future sequences from *Z*. In particular, we will use recurrent neural networks, which are known to provide good models of sequences from a wide range of domains; our method will also be applicable to other types of function approximators.

This will have two main advantages, addressing the limitations of OCF: first, unlike OCF, function approximators can discover generalizations across similar sequences, enabling the method to calculate good codes *Z* even for past sequences that were not seen previously. This is of paramount importance in settings where the state space is large, such as the set of words of a natural language. Second, the cost of storing and evaluating the function approximators will scale *linearly* with the length of observed sequences both in space and in time, as opposed to the exponential memory demand of OCF. This is crucial for modeling long dependencies.

### 4.1. Main Result: Variational Bound on Predictive Rate–Distortion

We will first describe the general method, without committing to a specific framework for function approximation yet. We will construct a bound on Predictive Rate–Distortion and optimize this bound in a parametric family of function approximators to obtain an encoding *Z* that is close to optimal for the nonparametric objective (Equation 7).

As in OCF (Section 3), we assume that a set of finite sample trajectories $x_{-M}\ldots x_{M-1}$ is available, and we aim to compress pasts of length *M* to predict futures of length *M*. To carry this out, we restrict the PRD objective (Equation 7) to such finite-length pasts and futures: (10)$$\max_{Z:Z⊥\overset{\to M}{X}|\overset{M\leftarrow }{X}}\left(I\left[Z,\overset{\to M}{X}\right]-\lambda \cdot I\left[\overset{M\leftarrow }{X},Z\right]\right).$$

It will be convenient to equivalently rewrite (Equation 10) as
(11)$$\min_{Z:Z⊥\overset{\to M}{X}|\overset{M\leftarrow }{X}}\left[H\left[\overset{\to M}{X}|Z\right]+\lambda \cdot I\left[\overset{M\leftarrow }{X},Z\right]\right],$$
where $H[\overset{\to M}{X}|Z]$ is the prediction loss. Note that minimizing prediction loss is equivalent to maximizing predictiveness $I[\overset{\to M}{X},Z]$.

When deploying such a predictive code *Z*, two components have to be computed: a distribution $P(Z|\overset{M\leftarrow }{X})$ that encodes past observations into a code *Z*, and a distribution $P(\overset{\to M}{X}|Z)$ that decodes the code *Z* into predictions about the future.

Let us assume that we already have some encoding distribution
(12)$$Z∼P_{\phi }\left(Z|\overset{M\leftarrow }{X}\right),$$
where ϕ is the encoder, expressed in some family of function approximators. The encoder transduces an observation sequence $\overset{M\leftarrow }{X}$ into the parameters of the distribution $P_{\phi }\left(\cdot |\overset{M\leftarrow }{X}\right)$. From this encoding distribution, one can obtain the optimal decoding distribution over future observations via Bayes’ rule: (13)$$\begin{array}{cc} P(\overset{\to M}{X}|Z) & =\frac{P(\overset{\to M}{X},Z)}{P(Z)}=\frac{E_{\overset{M\leftarrow }{X}}P\left(\overset{\to M}{X},Z|\overset{M\leftarrow }{X}\right)}{E_{\overset{M\leftarrow }{X}}P_{\phi }\left(Z|\overset{M\leftarrow }{X}\right)}{=}^{\left(*\right)}\frac{E_{\overset{M\leftarrow }{X}}P\left(\overset{\to M}{X}|\overset{M\leftarrow }{X}\right)P_{\phi }\left(Z|\overset{M\leftarrow }{X}\right)}{E_{\overset{M\leftarrow }{X}}P_{\phi }\left(Z|\overset{M\leftarrow }{X}\right)}, \end{array}$$
where (∗) uses the Markov condition $Z⊥\overset{\to M}{X}|\overset{M\leftarrow }{X}$. However, neither of the two expectations in the last expression of (Equation 13) is tractable, as they require summation over exponentially many sequences, and algorithms (e.g., dynamic programming) to compute this sum efficiently are not available in general. For a similar reason, the rate $I[\overset{M\leftarrow }{X},Z]$ of the code $Z∼P_{\phi }\left(Z|\overset{M\leftarrow }{X}\right)$ is also generally intractable.

Our method will be to introduce additional functions, also expressed using function approximators that approximate some of these intractable quantities: first, we will use a parameterized probability distribution *q* as an approximation to the intractable marginal $P\left(Z\right)=E_{\overset{M\leftarrow }{X}}P_{\phi }\left(Z|\overset{M\leftarrow }{X}\right)$: (14)$$q\left(Z\right)\;\mathrm{approximates}\;P\left(Z\right)=E_{\overset{M\leftarrow }{X}}P_{\phi }\left(Z|\overset{M\leftarrow }{X}\right).$$

Second, to approximate the decoding distribution $P(\overset{\to M}{X}|Z)$, we introduce a parameterized decoder ψ that maps points $Z\in R^{N}$ into probability distributions $P_{\psi }\left(\overset{\to M}{X}|Z\right)$ over future observations $\overset{\to M}{X}$: (15)$$P_{\psi }\left(\overset{\to M}{X}|Z\right)\;\mathrm{approximates}\;P\left(\overset{\to M}{X}|Z\right)$$
for each code $Z\in R^{N}$. Crucially, $P_{\psi }\left(\overset{\to M}{X}|Z\right)$ will be easy to compute efficiently, unlike the exact decoding distribution $P(\overset{\to M}{X}|Z)$.

If we fix a stochastic process ${\left(X_{t}\right)}_{t\in Z}$ and an encoder ϕ, then the following two bounds hold for *any* choice of the decoder ψ and the distribution *q*:

**Proposition** **1.**
*The loss incurred when predicting the future from Z via ψ upper-bounds the true conditional entropy of the future given Z, when predicting using the exact decoding distribution (Equation 13):*
(16)$$-E_{X}E_{Z∼\phi (X)}\left[\log P_{\psi }\left(\overset{\to M}{X}|Z\right)\right]\geq H\left[\overset{\to M}{X}|Z\right].$$

*Furthermore, equality is attained if and only if $P_{\psi }\left(\overset{\to M}{X}|Z\right)=P\left(\overset{\to M}{X}|Z\right)$.*


**Proof.** By Gibbs’ inequality:
$$\begin{array}{cc} -E_{X}E_{Z∼\phi (X)}\left[\log P_{\psi }\left(\overset{\to M}{X}|Z\right)\right] & \geq -E_{X}E_{Z∼\phi (X)}\left[\log P(\overset{\to M}{X}|Z)\right]\\ & =H[\overset{\to M}{X}|Z]. \end{array}$$ □

**Proposition** **2.**
*The KL Divergence between $P_{\phi }\left(Z|\overset{M\leftarrow }{X}\right)$ and q(Z), averaged over pasts $\overset{M\leftarrow }{X}$, upper-bounds the rate of Z:*
(17)$$\begin{array}{cc} E_{\overset{M\leftarrow }{X}}\left[D_{\mathrm{KL}}(P_{\phi }\left(Z|\overset{M\leftarrow }{X}\right)\|q(Z))\right] & =E_{\overset{M\leftarrow }{X}}E_{Z|\overset{M\leftarrow }{X}}\log\frac{P_{\phi }\left(Z|\overset{M\leftarrow }{X}\right)}{q(Z)}\\ & \geq E_{\overset{M\leftarrow }{X}}E_{Z|\overset{M\leftarrow }{X}}\log\frac{P_{\phi }\left(Z|\overset{M\leftarrow }{X}\right)}{P(Z)}\\ & =I[\overset{M\leftarrow }{X},Z]. \end{array}$$

*Equality is attained if and only if q(Z) is equal to the true marginal $P\left(Z\right)=E_{\overset{M\leftarrow }{X}}P_{\phi }\left(Z|\overset{M\leftarrow }{X}\right)$.*


**Proof.** The two equalities follow from the definition of KL Divergence and Mutual Information. To show the inequality, we again use Gibbs’ inequality:
$$\begin{array}{c} -E_{\overset{M\leftarrow }{X}}E_{Z|\overset{M\leftarrow }{X}}\log q\left(Z\right)=-E_{Z}\log q\left(Z\right)\geq -E_{Z}\log P\left(Z\right)=-E_{\overset{M\leftarrow }{X}}E_{Z|\overset{M\leftarrow }{X}}\log P\left(Z\right). \end{array}$$Here, equality holds if and only if q(Z)=P(Z), proving the second assertion. □

We now use the two propositions to rewrite the Predictive Rate–Distortion objective (Equation 18) in a way amenable to using function approximators, which is our main theoretical result, and the foundation of our proposed estimation method:

**Corollary** **1** (Main Result)**.**
*The Predictive Rate–Distortion objective (Equation 18)*
(18)$$\min_{Z:Z⊥\overset{\to M}{X}|\overset{M\leftarrow }{X}}\left[H\left[\overset{\to M}{X}|Z\right]+\lambda I\left[\overset{M\leftarrow }{X},Z\right]\right]$$
*is equivalent to*
(19)$$\min_{\phi ,\psi ,q}\left[E_{\overset{M\leftarrow }{X},\overset{\to M}{X}}\left[-E_{Z∼\phi (\overset{M\leftarrow }{X})}\left[\log P_{\psi }\left(\overset{\to M}{X}|Z\right)\right]+\lambda \cdot D_{\mathrm{KL}}\left[(P_{\phi }\left(Z|\overset{M\leftarrow }{X}\right)\|q(Z))\right]\right]\right],$$
*where ϕ,ψ,q range over all triples of the appropriate types described above.*

*From a solution to (Equation 19), one obtains a solution to (Equation 18) by setting $Z∼P_{\phi }\left(\cdot |\overset{M\leftarrow }{X}\right)$. The rate of this code is given as follows:*
(20)$$I\left[Z,\overset{M\leftarrow }{X}\right]=E_{\overset{M\leftarrow }{X}}\left[D_{\mathrm{KL}}(P_{\phi }\left(Z|\overset{M\leftarrow }{X}\right)\|q(Z))\right]$$
*and the prediction loss is given by*
(21)$$H\left[\overset{\to M}{X}|Z\right]=-E_{X_{\overset{M\leftarrow }{X},\overset{\to M}{X}}}E_{Z∼P_{\phi }\left(\overset{M\leftarrow }{X}\right)}\left[\log P_{\psi }\left(\overset{\to M}{X}|Z\right)\right].$$


**Proof.** By the two propositions, the term inside the minimization in (Equation 19) is an upper bound on (Equation 18), and takes on that value if and only if $P_{\phi }\left(\cdot |\overset{M\leftarrow }{X}\right)$ equals the distribution of the *Z* optimizing (Equation 18), and ψ,q are as described in those propositions. □

Note that the right-hand sides of (Equation 20) and (Equation 21) can both be estimated efficiently using Monte Carlo samples from $Z∼P_{\phi }\left(\overset{M\leftarrow }{X}\right)$.

If ϕ,ψ,q are not exact solutions to (Equation 19), the two propositions guarantee that we still have bounds on rate and prediction loss for the code *Z* generated by ϕ: (22)$$I\left[Z,\overset{M\leftarrow }{X}\right]\leq E_{\overset{M\leftarrow }{X}}\left[D_{\mathrm{KL}}(P_{\phi }\left(Z|\overset{M\leftarrow }{X}\right)\|q(Z))\right],$$
(23)$$H\left[\overset{\to M}{X}|Z\right]\leq -E_{X_{\overset{M\leftarrow }{X},\overset{\to M}{X}}}E_{Z∼P_{\phi }\left(\overset{M\leftarrow }{X}\right)}\left[\log P_{\psi }\left(\overset{\to M}{X}|Z\right)\right].$$

To carry out the optimization (Equation 19), we will restrict ϕ,ψ,q to a powerful family of parametric families of function approximators, within which we will optimize the objective with gradient descent. While the solutions may not be exact solutions to the nonparametric objective (Equation 19), they will still satisfy the bounds (Equation 22) and (Equation 23), and—if the family of approximators is sufficiently rich—can come close to turning these into the equalities (Equation 20) and (Equation 21).

### 4.2. Choosing Approximating Families

For our method of Neural Predictive Rate–Distortion (NPRD), we choose the approximating families for the encoder ϕ, the decoder ψ, and the distribution *q* to be certain types of neural networks that are known to provide strong and general models of sequences and distributions.

For ϕ and ψ, we use recurrent neural networks with Long Short Term Memory (LSTM) cells [32], widely used for modeling sequential data across different domains. We parameterize the distribution $P_{\phi }\left(Z|\overset{M\leftarrow }{X}\right)$ as a Gaussian whose mean and variance are computed from the past $\overset{M\leftarrow }{X}$: We use an LSTM network to compute a vector $h\in R^{k}$ from the past observations $\overset{M\leftarrow }{X}$, and then compute
(24)$$Z∼N(W_{\mu }h,{\left(W_{\sigma }h\right)}^{2}I_{k\times k}),$$
where $W_{\mu },W_{\sigma }\in R^{k\times k}$ are parameters. While we found Gaussians sufficiently flexible for ϕ, more powerful encoders could be constructed using more flexible parametric families, such as normalizing flows [19,33].

For the decoder ψ, we use a second LSTM network to compute a sequence of vector representations $g_{t}=\psi \left(Z,X_{0\ldots t-1}\right)$ ($g_{t}\in R^{k}$) for t=0,…M−1. We compute predictions using the softmax rule
(25)$$P_{\psi }\left(X_{t}=s_{i}|X_{1\ldots t-1},Z\right)\propto \exp\left({\left(W_{o}g_{t}\right)}_{i}\right)$$
for each element $s_{i}$ of the state space $S=\{s_{1},\ldots ,s_{\left|S\right|}\}$, and $W_{o}\in R^{|S|\times k}$ is a parameter matrix to be optimized together with the parameters of the LSTM networks.

For *q*, we choose the family of Neural Autoregressive Flows [20]. This is a parametric family of distributions that allows efficient estimation of the probability density and its gradients. This method widely generalizes a family of prior methods [19,33,34], offering efficient estimation while surpassing prior methods in expressivity.

### 4.3. Parameter Estimation and Evaluation

We optimize the parameters of the neural networks expressing ϕ,ψ,q for the objective (Equation 19) using Backpropagation and Adam [35], a standard and widely used gradient descent-based method for optimizing neural networks. During optimization, we approximate the gradient by taking a single sample from *Z* (Equation 24) per sample trajectory $X_{-M},\ldots ,X_{M-1}$ and use the reparametrized gradient estimator introduced by Kingma and Welling [36]. This results in an unbiased estimator of the gradient of (Equation 19) w.r.t. the parameters of ϕ,ψ,q.

Following standard practice in machine learning, we split the data set of sample time series into three partitions (training set, validation set, and test set). We use the training set for optimizing the parameters as described above. After every pass through the training set, the objective (Equation 19) is evaluated on the validation set using a Monte Carlo estimate with one sample *Z* per trajectory; optimization terminates once the value on the validation set does not decrease any more.

After optimizing the parameters on a set of observed trajectories, we estimate rate and prediction loss on the test set. Given parameters for ϕ,ψ,q, we evaluate the PRD objective (Equation 19), rate (Equation 22), and the prediction loss (Equation 23) on the test set by taking, for each time series $X_{-M}\ldots X_{M-1}=\overset{M\leftarrow }{X}\overset{\to M}{X}$ from the test set, a single sample $z∼N(\mu ,\sigma^{2})$ and computing Monte Carlo estimates for rate
(26)$$E_{X}\left[D_{\mathrm{KL}}(P_{\phi }\left(Z|\overset{M\leftarrow }{X}\right)\|q(Z))\right]\approx \frac{1}{N}\sum_{X_{-M\ldots M}\in TestData}\log\frac{p_{N(\mu ,\sigma^{2})}\left(z\right)}{q(z)},$$
where $p_{N(\mu ,\sigma^{2})}\left(z\right)$ is the Gaussian density with $\mu ,\sigma^{2}$ computed from $\overset{M\leftarrow }{X}$ as in (Equation 24), and prediction loss
(27)$$-E_{Z∼\phi (\overset{M\leftarrow }{X})}\left[\log P_{\psi }\left(\overset{\to M}{X}|Z\right)\right]\approx -\frac{1}{N}\sum_{X_{-M\ldots M}\in TestData}\log P_{\psi }\left(\overset{\to M}{X}|Z\right).$$

Thanks to (Equation 22) and (Equation 23), these estimates are guaranteed to be *upper bounds* on the true rate and prediction loss achieved by the code $Z∼N(\mu ,\sigma^{2})$, up to sampling error introduced into the Monte Carlo estimation by sampling *z* and the finite size of the test set.

It is important to note that this sampling error is different from the overfitting issue affecting OCF: Our Equations (Equation 26) and (Equation 27) provide *unbiased* estimators of upper bounds, whereas overfitting *biases* the values obtained by OCF. Given that Neural Predictive Rate–Distortion provably provide upper bounds on rate and prediction loss (up to sampling error), one can objectively compare the quality of different estimation methods: among methods that provide upper bounds, the one that provides the lowest such bound for a given problem is the one giving results closest to the true curve.

#### Estimating Predictiveness

Given the estimate for prediction loss, we estimate predictiveness $I[Z,\overset{\to M}{X}]$ with the following method. We use the encoder network that computes the code vector *h* (Equation 24) to also estimate the marginal probability of the past observation sequence, $P_{\eta }\left(\overset{M\leftarrow }{X}\right)$. $P_{\eta }$ has support over sequences of length *M*. Similar to the decoder ψ, we use the vector representations $f_{t}\in R^{k}$ computed by the LSTM encoder after processing $X_{-M\ldots t}$ for t=−M,…,−1, and then compute predictions using the softmax rule
(28)$$P_{\eta }\left(X_{t}=s_{i}|X_{1\ldots t-1},Z\right)\propto \exp\left({\left(W_{o^{\prime }}f_{t}\right)}_{i}\right),$$
where $W_{o^{\prime }}\in R^{|S|\times k}$ is another parameter matrix.

Because we consider stationary processes, we have that the cross-entropy under $P_{\eta }$ of $\overset{\to M}{X}$ is equal to the cross-entropy of $\overset{M\leftarrow }{X}$ under the same encoding distribution: $E_{X_{\overset{M\leftarrow }{X}\overset{\to M}{X}}}\left[\log P_{\eta }\left(\overset{\to M}{X}\right)\right]=-E_{X_{\overset{M\leftarrow }{X}\overset{\to M}{X}}}\left[\log P_{\eta }\left(\overset{M\leftarrow }{X}\right)\right]$. Using this observation, we estimate the predictiveness $I\left[Z,\overset{\to M}{X}\right]=H\left[\overset{\to M}{X}\right]-H\left[\overset{\to M}{X}|Z\right]$ by the difference between the corresponding cross-entropies on the test set [37]: (29)$$-E_{X_{\overset{M\leftarrow }{X}\overset{\to M}{X}}}\left[\log P_{\eta }\left(\overset{\to M}{X}\right)-\log P_{\psi }\left(\overset{\to M}{X}|Z\right)\right],$$
which we approximate using Monte Carlo sampling on the test set as in (Equation 26) and (Equation 27).

In order to optimize parameters for estimation of $P_{\eta }$, we add the cross-entropy term $-E_{X_{\overset{M\leftarrow }{X}\overset{\to M}{X}}}\left[\log P_{\eta }\left(\overset{M\leftarrow }{X}\right)\right]$ to the PRD objective (Equation 19) during optimization, so that the full training objective comes out to: (30)$$\min_{\phi ,\psi ,q,\eta }\left[E_{\overset{M\leftarrow }{X},\overset{\to M}{X}}\left[-E_{Z∼\phi (\overset{M\leftarrow }{X})}\left[\log P_{\psi }\left(\overset{\to M}{X}|Z\right)\right]+\lambda \cdot D_{\mathrm{KL}}\left[(P_{\phi }\left(Z|\overset{M\leftarrow }{X}\right)\|q(Z))\right]-\log P_{\eta }\left(\overset{M\leftarrow }{X}\right)\right]\right].$$

Again, by Gibbs’ inequality and Propositions 1 and 2, this is minimized when $P_{\eta }$ represents the true distribution over length-*M* blocks $P(\overset{M\leftarrow }{X})$, $P_{\phi }\left(Z|\overset{M\leftarrow }{X}\right)$ describes an optimal code for the given λ, *q* is its marginal distribution, and $P_{\psi }\left(\overset{\to M}{X}|Z\right)$ is the Bayes-optimal decoder. For approximate solutions to this augmented objective, the inequalities (Equation 22) and (Equation 23) will also remain true due to Propositions 1 and 2.

### 4.4. Related Work

In (Equation 19), we derived a variational formulation of Predictive Rate–Distortion. This is formally related to a variational formulation of the Information Bottleneck that was introduced by [38], who applied it to neural-network based image recognition. Unlike our approach, they used a fixed diagonal Gaussian instead of a flexible parametrized distribution for *q*. Some recent work has applied similar approaches to the modeling of sequences, employing models corresponding to the objective (Equation 19) with λ=1 [39,40,41,42].

In the neural networks literature, the most commonly used method using variational bounds similar to Equation (Equation 19) is the Variational Autoencoder [36,43], which corresponds to the setting where λ=1 and the predicted output is equal to the observed input. The β-VAE [44], a variant of the Variational Autoencoder, uses λ>1 (whereas the Predictive Rate–Distortion objective (Equation 7) uses λ∈(0,1)), and has been linked to the Information Bottleneck by [45].

## 5. Experiments

We now test the ability of our new method NPRD to estimate rate–distortion curves. Before we apply NPRD to obtain the first estimates of Predictive Rate–Distortion for natural language in Section 6, we validate the method on processes whose trade-off curves are analytically known, and compare with OCF.

### 5.1. Implementation Details

#### OCF

As discussed in Section 3, OCF is affected by overfitting, and will systematically overestimate the predictiveness achieved at a given rate [6]. To address this problem, we follow the evaluation method used for NPRD, evaluating rate and predictiveness on held-out test data. We partition the available time series data into a training and test set. We use the training set to create the encoder $P(Z|\overset{M\leftarrow }{X})$ using the Blahut–Arimoto algorithm as described by Still et al. [6]. We then use the held-out test set to estimate rate and prediction loss. This method not only enables fair comparison between OCF and NPRD, but also provides a more realistic evaluation, by focusing on the performance of the code *Z* when deployed on new data samples. For rate, we use the same variational bound that we use for NPRD, stated in Proposition 2: (31)$$\frac{1}{N}\sum_{\overset{M\leftarrow }{X}\in \mathrm{Test}\;\mathrm{Data}}D_{\mathrm{KL}}(P(Z|\overset{M\leftarrow }{X})\|s(Z)),$$
where $P(Z|\overset{M\leftarrow }{X})$ is the encoder created by the Blahut–Arimoto algorithm, and s(Z) is the marginal distribution of *Z* on the training set. *N* is the number of sample time series in the test data. In the limit of enough training data, when s(Z) matches the actual population marginal of *Z*, (Equation 31) is an unbiased estimate of the rate. We estimate the prediction loss on the future observations as the empirical cross-entropy, i.e., the variational bound stated in Proposition 1: (32)$$\frac{1}{N}\sum_{\overset{M\leftarrow }{X},\overset{\to M}{X}\in \mathrm{Test}\;\mathrm{Data}}E_{Z∼P(\cdot |\overset{M\leftarrow }{X})}\log P\left(\overset{\to M}{X}|Z\right),$$
where $P(\overset{\to M}{X}|Z)$ is the decoder obtained from the Blahut–Arimoto algorithm on the training set. Thanks to Propositions 1 and 2, these quantities provide upper bounds, up to sampling error introduced by finiteness of the held-out data. Again, sampling error does not bias the results in either direction, unlike overfitting, which introduces a systematic bias.

Held-out test data may contain sequences that did not occur in the training data. Therefore, we add a pseudo-sequence ω and add pseudo-observations $\left(\omega ,\overset{\to M}{X}\right)$, $\left(\overset{M\leftarrow }{X},\omega \right)$, (ω,ω) for all observed sequences $\overset{M\leftarrow }{X}\overset{\to M}{X}$ to the matrix of observed counts that serves as the input to the Blahut–Arimoto algorithm. These pseudo-observations were assigned pseudo-counts γ in this matrix of observed counts; we found that a wide range of values ranging from 0.0001 to 1.0 yielded essentially the same results. When evaluating the codebook on held-out data, previously unseen sequences were mapped to ω.

#### Neural Predictive Rate–Distortion

For all experiments, we used M=15. Neural networks have hyperparameters, such as the number of units and the step size used in optimization, which affect the quality of approximation depending on properties of the dataset and the function being approximated. Given that NPRD provably provides upper bounds on the PRD objective (Equation 18), one can in principle identify the best hyperparameters for a given process by choosing the combination that leads to the lowest estimated upper bounds. As a computationally more efficient method, we defined a range of plausible hyperparameters based both on experience reported in the literature, and considerations of computational efficiency. These parameters are discussed in Appendix A. We then randomly sampled, for each of the processes that we experimented on, combinations of λ and these hyperparameters to run NPRD on. We implemented the model using PyTorch [46].

### 5.2. Analytically Tractable Problems

We first test NPRD on two processes where the Predictive Rate–Distortion trade-off is analytically tractable. The Even Process [2] is the process of 0/1 IID coin flips, conditioned on all blocks of consecutive ones having even length. Its complexity and excess entropy are both ≈0.63 nats. It has infinite Markov order, and Marzen and Crutchfield [2] find that OCF (at M=5) performs poorly. The true Predictive Rate–Distortion curve was computed in [2] using the analytically known ϵ-machine. The Random Insertion Process [2] consists of sequences of uniform coin flips $X_{t}\in \left\{0,1\right\}$, subject to the constraint that, if $X_{t-2}$ was a 0, then $X_{t}$ has to be a 1.

We applied NPRD to these processes by training on 3M random trajectories of length 30, and using 3000 additional trajectories for validation and test data. For each process, we ran NPRD 1000 times for random choices of λ∈[0,1]. Due to computational constraints, when running OCF, we limited sample size to 3000 trajectories for estimation and as held-out data. Following Marzen and Crutchfield [2], we ran OCF for M=1,…,5.

The resulting estimates are shown in Figure 1, together with the analytical rate–distortion curves computed by Marzen and Crutchfield [2]. Individual runs of NPRD show variation (red dots), but most runs lead to results close to the analytical curve (gray line), and strongly surpass the curves computed by OCF at M=5. Bounding the trade-off curve using the sets of runs of NPRD results in a close fit (red line) to the analytical trade-off curves.

#### Recovering Causal States

Does NPRD lead to interpretable codes *Z*? To answer this, we further investigated the NPRD approximation to the Random Insertion Process (RIP), obtained in the previous paragraph. The ϵ-machine was computed by Marzen and Crutchfield [2] and is given in Figure 2 (left). The process has three causal states: State *A* represents those pasts where the future starts with $1^{2k}0$ (k=0,1,2,…)—these are the pasts ending in either 001 or $10111^{m}$ (m=0,1,2,…). State *B* represents those pasts ending in 10—the future has to start with 01 or 11. State *C* represents those pasts ending in either 00 or 01—the future has to start with $1^{2k+1}0$ (k=0,1,2,…).

The analytical solution to the Predictive Rate–Distortion problem was computed by Marzen and Crutchfield [2]. At λ>0.5, the optimal solution collapses *A* and *B* into a single codeword, while all three states are mapped to separate codewords for λ≤0.5.

Does NPRD recover this picture? We applied PCA to samples from *Z* computed at two different values of λ, λ=0.25 and λ=0.6. The first two principal components of *Z* are shown in Figure 3. Samples are colored by the causal states corresponding to the pasts of the trajectories that were encoded into the respective points by NPRD. On the left, obtained at λ=0.6, the states A and B are collapsed, as expected. On the right, obtained at λ=0.25, the three causal states are reflected as distinct modes of *Z*. Note that, at finite *M*, a fraction of pasts is ambiguous between the green and blue causal states; these are colored in black and NPRD maps them into a region between the modes corresponding to these states.

In Figure 2 (right), we record, for each of the three modes, to which cluster the distribution of the code *Z* shifts when a symbol is appended. We restrict to those strings that have nonzero probability for RIP (no code will ever be needed for other strings). For comparison, we show the ϵ-machine computed by Marzen and Crutchfield [2]. Comparing the two diagrams shows that NPRD effectively recovers the ϵ-machine: the three causal states are represented by the three different modes of *Z*, and the effect of appending a symbol also mirrors the state transitions of the ϵ-machine.

#### A Process with Many Causal States

We have seen that NPRD recovers the correct trade-off, and the structure of the causal states, in processes with a small number of causal states. How does it behave when the number of causal states is very large? In particular, is it capable of extrapolating to causal states that were never seen during training?

We consider the following process, which we will call Copy3: $X_{-15},\ldots ,X_{-1}$, are independent uniform draws from {1,2,3}, and $X_{1}=X_{-1},\ldots ,X_{15}=X_{-15}$. This process deviates a bit from our usual setup since we defined it only for t∈{−15,…,15}, but it is well-suited to investigating this question: the number of causal states is $3^{15}\approx 14$ million. With exactly the same setup as for the Even and RIP processes, NPRD achieved essentially zero distortion on unseen data, even though the number of training samples (3 Million) was far lower than the number of distinct causal states. However, we found that, in this setup, NPRD overestimated the rate. Increasing the number of training samples from 3M to 6M, NPRD recovered codebooks that achieved both almost zero distortion and almost optimal rate, on fresh samples (Figure 4). Even then, the number of distinct causal states is more than twice the number of training samples. These results demonstrate that, by using function approximation, NPRD is capable of extrapolating to unseen causal states, encoding and decoding appropriate codes on the fly.

Note that one could easily design an optimal decoder and encoder for Copy3 by hand—the point of this experiment is to demonstrate that NPRD is capable of inducing such a codebook purely from data, in a general-purpose, off-the-shelf manner. This contrasts with OCF: without optimizations specific to the task at hand, a direct application of OCF would require brute-force storing of all 14 million distinct pasts and futures.

## 6. Estimating Predictive Rate–Distortion for Natural Language

We consider the problem of estimating rate–distortion for natural language. Natural language has been a testing ground for information-theoretic ideas since Shannon’s work. Much interest has been devoted to estimating the entropy rate of natural language [10,47,48,49]. Indeed, the information density of language has been linked to human processing effort and to language structure. The word-by-word information content has been shown to impact human processing effort as measured both by per-word reading times [50,51,52] and by brain signals measured through EEG [53,54]. Consequently, prediction is a fundamental component across theories of human language processing [54]. Relatedly, the Uniform Information Density and Constant Entropy Rate hypotheses [55,56,57] state that languages order information in sentences and discourse so that the entropy rate stays approximately constant.

The relevance of prediction to human language processing makes the *difficulty* of prediction another interesting aspect of language complexity: Predictive Rate–Distortion describes how much memory of the past humans need to maintain to predict future words accurately. Beyond the entropy rate, it thus forms another important aspect of linguistic complexity.

Understanding the complexity of prediction in language holds promise for enabling a deeper understanding of the nature of language as a stochastic process, and to human language processing. Long-range correlations in text have been a subject of study for a while [58,59,60,61,62,63]. Recently, Dębowski [64] has studied the excess entropy of language across long-range discourses, aiming to better understand the nature of the stochastic processes underlying language. Koplenig et al. [65] shows a link between traditional linguistic notions of grammatical structure and the information contained in word forms and word order. The idea that predicting future words creates a need to represent the past well also forms a cornerstone of theories of how humans process sentences [66,67].

We study prediction in the range of the words in individual sentences. As in the previous experiments, we limit our computations to sequences of length 30, already improving over OCF by an order of magnitude. One motivation is that, when directly estimating PRD, computational cost has to increase with the length of sequences considered, making the consideration of sequences of hundreds or thousands of words computationally infeasible. Another motivation for this is that we are ultimately interested in Predictive Rate–Distortion as a model of memory in human processing of grammatical structure, formalizing psycholinguistic models of how humans process individual sentences [66,67], and linking to studies of the relation between information theory and grammar [65].

### 6.1. Part-of-Speech-Level Language Modeling

We first consider the problem of predicting English on the level of Part-of-Speech (POS) tags, using the Universal POS tagset [68]. This is a simplified setting where the vocabulary is small (20 word types), and one can hope that OCF will produce reasonable results. We use the English portions of the Universal Dependencies Project [69] tagged with Universal POS Tags [68], consisting of approximately 586 K words. We used the training portions to estimate NPRD and OCF, and the validation portions to estimate the rate–distortion curve. We used NPRD to generate 350 codebooks for values of λ sampled from [0, 0.4]. We were only able to run OCF for M≤3, as the number of sequences exceeds $10^{4}$ already at M=4.

The PRD curve is shown in Figure 5 (left). In the setting of low rate and high distortion, NPRD and OCF (blue, M=1,2,3) show essentially identical results. This holds true until $I[Z,\vec{X}]\approx 0.7$, at which point the bounds provided by OCF deteriorate, showing the effects of overfitting. NPRD continues to provide estimates at greater rates.

Figure 5 (center) shows rate as a function of $\log\frac{1}{\lambda }$. Recall that λ is the trade-off-parameter from the objective function (Equation 7). In Figure 5 (right), we show rate and the mutual information with the future, as a function of $\log\frac{1}{\lambda }$. As λ→0, NPRD (red, M=15) continues to discover structure, while OCF (blue, plotted for M=1,2,3) exhausts its capacity.

Note that NPRD reports rates of 15 nats and more when modeling with very low distortion. A discrete codebook would need over 3 million distinct codewords for a code of such a rate, exceeding the size of the training corpus (about 500 K words), replicating what we found for the Copy3 process: Neural encoders and decoders can use the geometric structure of the code space to encode generalizations across different dimensions, supporting a very large effective number of distinct possible codes. Unlike discrete codebooks, the geometric structure makes it possible for neural encoders to construct appropriate codes ’on the fly’ on new input.

### 6.2. Discussion

Let us now consider the curves in Figure 5 in more detail. Fitting parametric curves to the empirical PRD curves in Figure 5, we find a surprising result that the statistical complexity of English sentences at the POS level appears to be unbounded.

The rate-predictiveness curve (left) shows that, at low rates, predictiveness is approximately proportional to the rate. At greater degrees of predictiveness, the rate grows faster and faster, whereas predictiveness seems to asymptote to ≈1.1 nats. The asymptote of predictiveness can be identified with the mutual information between past and future observations, $E_{0}:=I\left[\overset{M\leftarrow }{X},\overset{\to M}{X}\right]$, which is a lower bound on the excess entropy. The rate should asymptote to the statistical complexity. Judging by the curve, natural language—at the time scale we are measuring in this experiment—has a statistical complexity much higher than its excess entropy: at the highest rate measured by NPRD in our experiment, rate is about 20 nats, whereas predictiveness is about 1.1 nats. If these values are correct, then—due to the convexity of the rate-predictivity curve—statistical complexity exceeds the excess entropy by a factor of at least $\frac{20}{1.1}$. Note that this picture agrees qualitatively with the OCF results, which suggest a lower-bound on the ratio of at least $\frac{2.5}{0.6}>5$.

Now, turning to the other plots in Figure 5, we observe that rate increases at least linearly with $\log\frac{1}{\lambda }$, whereas predictiveness again asymptotes. This is in qualitative agreement with the picture gained from the rate-predictiveness curve.

Let us consider this more quantitatively. Based on Figure 5 (center), we make the ansatz that the map from $\log\frac{1}{\lambda }$ to the rate $R:=I[\overset{M\leftarrow }{X},Z]$ is superlinear: (33)$$R=\alpha {\left(\log\frac{1}{\lambda }\right)}^{\beta },$$
with α>0,β>1. We fitted $R\approx {\left(\log\frac{1}{\lambda }\right)}^{1.7}$ (α=1, β=1.7). Equivalently,
(34)$$\frac{1}{\lambda }=\exp\left(\frac{1}{\alpha^{1/\beta }}R^{1/\beta }\right).$$

From this, we can derive expressions for rate $R:=I[\overset{M\leftarrow }{X},Z]$ and predictiveness $P:=I[Z,\overset{\to M}{X}]$ as follows. For the solution of Predictive Rate–Distortion (Equation 10), we have
(35)$$\frac{\partial P}{\partial \theta }-\lambda \frac{\partial R}{\partial \theta }=0,$$
where θ is the codebook defining the encoding distribution $P(Z|\overset{M\leftarrow }{X})$, and thus
(36)$$\lambda =\frac{\partial P}{\partial R}.$$

Our ansatz therefore leads to the equation
(37)$$\frac{\partial P}{\partial R}=\exp\left(-\frac{1}{\alpha^{1/\beta }}R^{1/\beta }\right).$$

Qualitatively, this says that predictiveness *P* asymptotes to a finite value, whereas rate *R*—which should asymptote to the statistical complexity—is unbounded.

Equation (Equation 37) has the solution
(38)$$P=C-\alpha \beta \cdot \Gamma \left(\beta ,{\left(R/\alpha \right)}^{1/\beta }\right),$$
where Γ is the incomplete Gamma function. Since $\lim_{R\to \infty }P=C$, the constant *C* has to equal the maximally possible predictiveness $E_{0}:=I\left[\overset{M\leftarrow }{X},\overset{\to M}{X}\right]$.

Given the values fitted above (α=1, β=1.7), we found that $E_{0}=1.13$ yielded a good fit. Using (Equation 33), this can be extended without further parameters to the third curve in Figure 5. Resulting fits are shown in Figure 6.

Note that there are other possible ways of fitting these curves; we have described a simple one that requires only two parameters α>0, β>1, in addition to a guess for the maximal predictiveness $E_{0}$. In any case, the results show that natural language shows an approximately linear growth of predictiveness with a rate at small rates, and exploding rates at diminishing returns in predictiveness later.

### 6.3. Word-Level Language Modeling

We applied NPRD to the problem of predicting English on the level of part-of-speech tags in Section 6.1. We found that the resulting curves were described well by Equation (Equation 37). We now consider the more realistic problem of prediction at the level of words, using data from multiple languages. This problem is much closer to the task faced by a human in the process of comprehending text, having to encode prior observations so as to minimize prediction loss on the upcoming words. We will examine whether Equation (Equation 37) describes the resulting trade-off in this more realistic setting, and whether it holds across languages.

For the setup, we followed a standard setup for recurrent neural language modeling. The hyperparameters are shown in Table A1. Following standard practice in neural language modeling, we restrict the observation space to the most frequent $10^{4}$ words; other words are replaced by their part-of-speech tag. We do this for simplicity and to stay close to standard practice in natural language processing; NPRD could deal with unbounded state spaces through a range of more sophisticated techniques such as subword modeling and character-level prediction [70,71].

We used data from five diverse languages. For English, we turn to the Wall Street Journal portion of the Penn Treebank [72], a standard benchmark for language modeling, containing about 1.2 million tokens. For Arabic, we pooled all relevant portions of the Universal Dependencies treebanks [73,74,75]. We obtained 1 million tokens. We applied the same method to construct a Russian corpus [76], obtaining 1.2 million tokens. For Chinese, we use the Chinese Dependency Treebank [77], containing 0.9 million tokens. For Japanese, we use the first 2 million words from a large processed corpus of Japanese business text [78]. For all these languages, we used the predefined splits into training, validation, and test sets.

For each language, we sampled about 120 values of $\log\frac{1}{\lambda }$ uniformly from [−6,0] and applied NPRD to these. The resulting curves are shown in Figure 7 and Figure 8, together with fitted curves resulting from Equation (Equation 37). As can be seen, the curves are qualitatively very similar across languages to what we observed in Figure 6: In all languages, rate initially scales linearly with predictiveness, but diverges as the predictiveness approaches its supremum $E_{0}$. As a function of $\log\frac{1}{\lambda }$, rate grows at a slightly superlinear speed, confirming our ansatz (Equation 33).

These results confirm our results from Section 6.1. At the time scale of individual sentences, Predictive Rate–Distortion of natural language appears to quantitatively follow Equation (Equation 37). NPRD reports rates up to ≈60 nats, more than ten times the largest values of predictiveness. On the other hand, the growth of rate with predictiveness is relatively gentle in the low-rate regime. We conclude that predicting words in natural language can be approximated with small memory capacity, but more accurate prediction requires very fine-grained memory representations.

### 6.4. General Discussion

Our analysis of PRD curves for natural language suggests that human language is characterized by very high and perhaps infinite statistical complexity, beyond its excess entropy. In a similar vein, Dębowski [64] has argued that the excess entropy of connected texts in natural language is infinite (in contrast, our result is for isolated sentences). If the statistical complexity of natural language is indeed infinite, then statistical complexity is not sufficiently fine-grained as a complexity metric for characterizing natural language.

We suggest that the PRD curve may form a more natural complexity metric for highly complex processes such as language. Among those processes with infinite statistical complexity, some will have a gentle PRD curve—meaning that they can be well-approximated at low rates—while others will have a steep curve, meaning they cannot be well-approximated at low rates. We conjecture that, although natural language may have infinite statistical complexity, it has a gentler PRD curve than other processes with this property, meaning that achieving a reasonable approximation of the predictive distribution does not require inordinate memory resources.

## 7. Conclusions

We introduce Neural Predictive Rate–Distortion (NPRD), a method for estimating Predictive Rate–Distortion when only sample trajectories are given. Unlike OCF, the most general prior method, NPRD scales to long sequences and large state spaces. On analytically tractable processes, we show that it closely fits the analytical rate–distortion curve and recovers the causal states of the process. On part-of-speech-level modeling of natural language, it agrees with OCF in the setting of low rate and short sequences; outside these settings, OCF fails due to combinatorial explosion and overfitting, while NPRD continues to provide estimates. Finally, we use NPRD to provide the first estimates of Predictive Rate–Distortion for modeling natural language in five different languages, finding qualitatively very similar curves in all languages.

All code for reproducing the results in this work is available at https://github.com/m-hahn/predictive-rate--distortion.

## Figures and Tables

**Figure 1 entropy-21-00640-f001:**
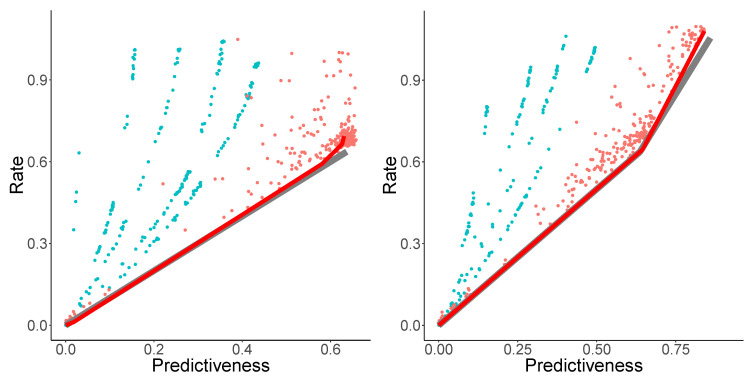
Rate–Distortion for the Even Process (**left**) and the Random Insertion Process (**right**). Gray lines: analytical curves; Red dots: multiple runs of NPRD; red line: trade-off curve computed from NPRD runs; blue: OCF for M≤5.

**Figure 2 entropy-21-00640-f002:**
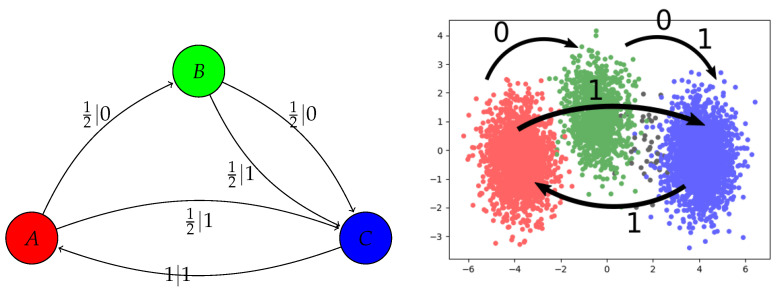
Recovering the ϵ-machine from NPRD. **Left**: The ϵ-machine of the Random Insertion Process, as described by [2]. **Right**: After computing a code *Z* from a past $x_{-15\ldots -1}$, we recorded which of the three clusters the code moves to when appending the symbol 0 or 1 to the past sequence. The resulting transitions mirror those in the ϵ-machine.

**Figure 3 entropy-21-00640-f003:**
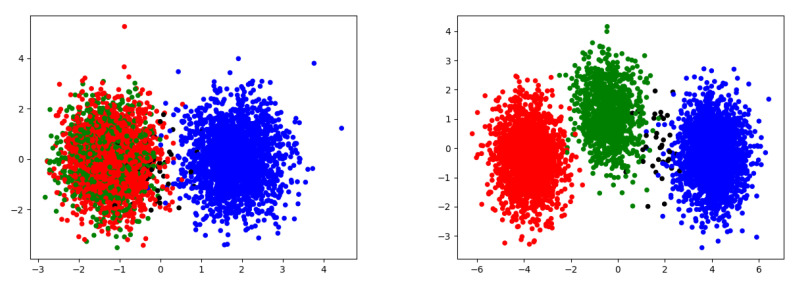
Applying Principal Component Analysis to 5000 sampled codes *Z* for the Random Insertion Process, at λ=0.6 (**left**) and λ=0.25 (**right**). We show the first two principal components. Samples are colored according to the states in the ϵ-machine. There is a small number of samples from sequences that, at M=15, cannot be uniquely attributed to any of the states (ambiguous between A and C); these are indicated in black.

**Figure 4 entropy-21-00640-f004:**
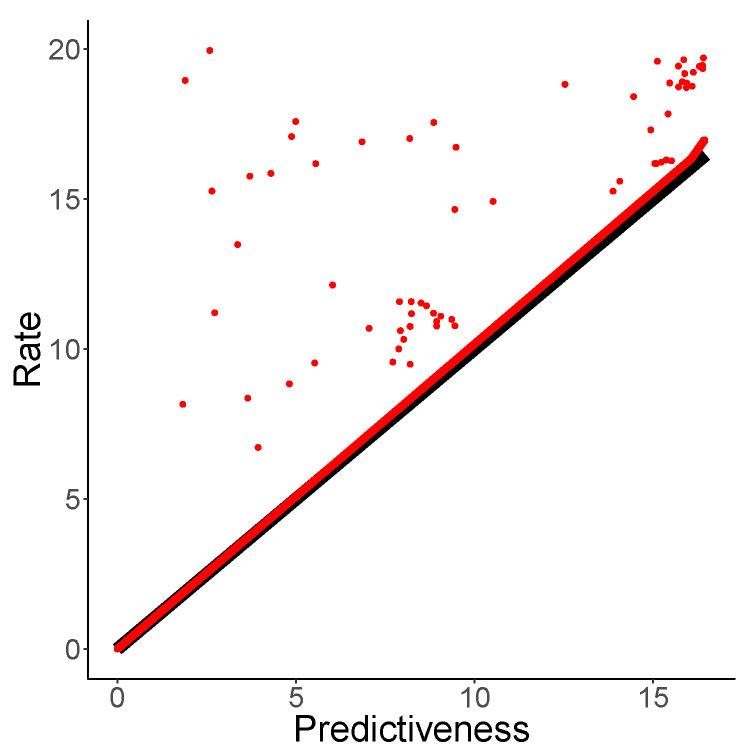
Rate–Distortion for the Copy3 process. We show NPRD samples, and the resulting upper bound in red. The gray line represents the anaytical curve.

**Figure 5 entropy-21-00640-f005:**
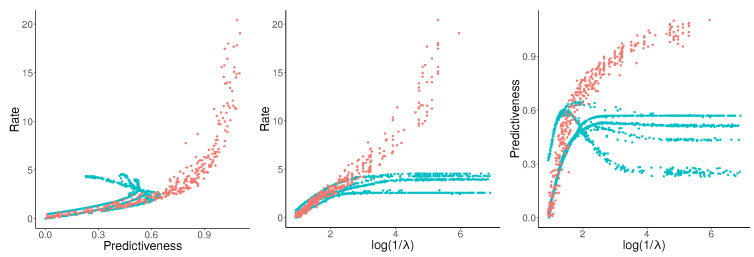
**Left**: Rate-Predictiveness for English POS modeling. Center and right: Rate (**Center**) and Predictiveness (**Right**) on English POS Modeling, as a function of −logλ. As λ→0, NPRD (red, M=15) continues to discover structure, while OCF (blue, plotted for M=1,2,3) exhausts its capacity.

**Figure 6 entropy-21-00640-f006:**
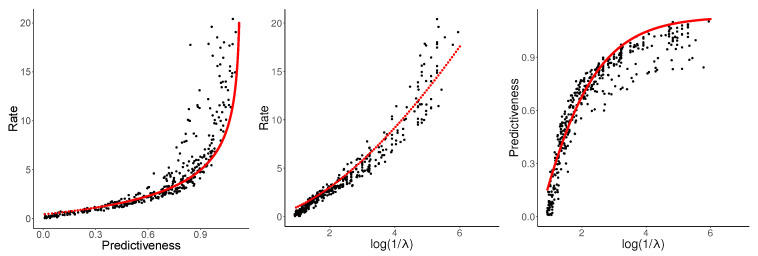
Interpolated values for POS-level prediction of English (compare Figure 5).

**Figure 7 entropy-21-00640-f007:**
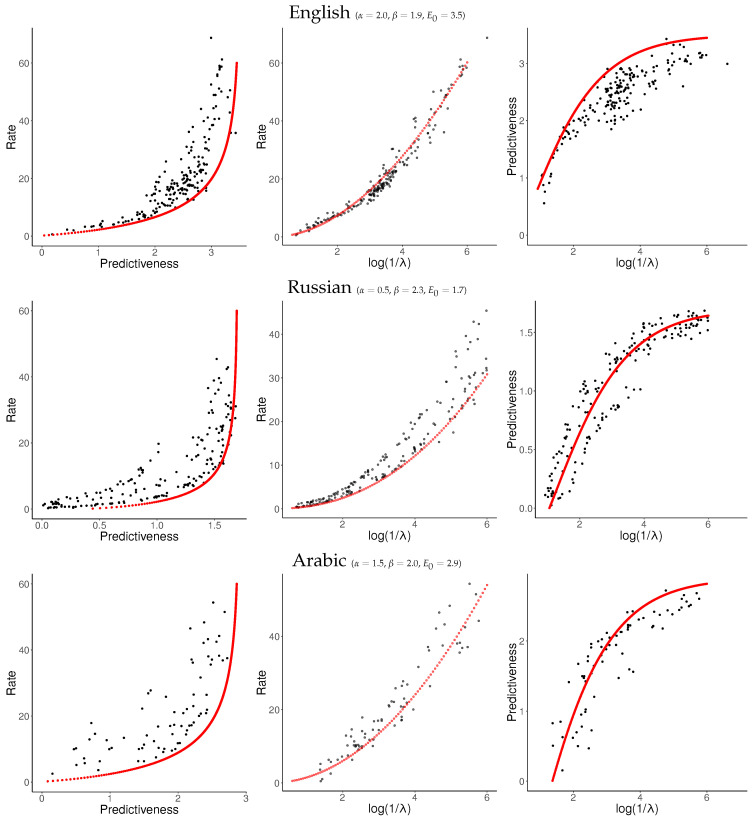
Word-level results.

**Figure 8 entropy-21-00640-f008:**
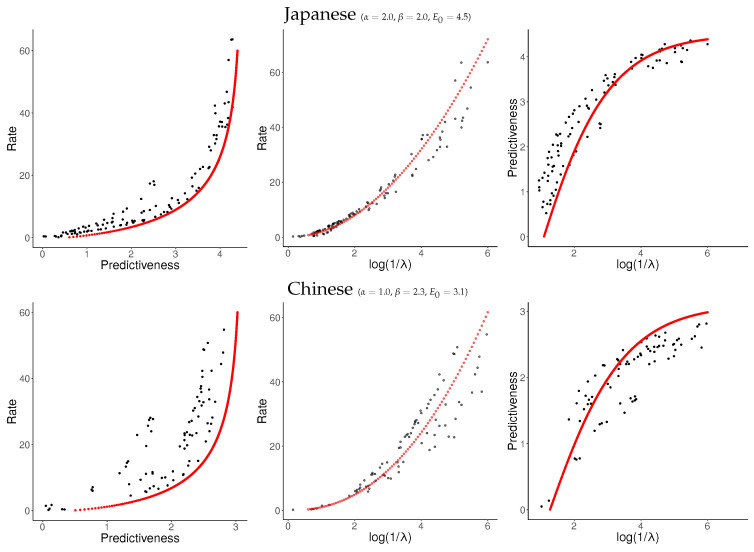
Word-level results (cont.).

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
