# Peer review of "Estimating Predictive Rate–Distortion Curves via Neural Variational Inference"

_entropy, 2019, doi:10.3390/e21070640_

Round 1
Reviewer 1 Report
This work proposes a neural network-based approach for tackling the past-future information bottleneck objective for complex discrete stochastic processes. All probability distributions involved in the objective are approximated by neural networks. To that end, an upper bound to the objective is derived, which the neural networks are trained to optimize. This approach is used to estimate rate-predictivity curves for multiple natural language corpora, as a more informative alternative to complexity, in the case where the latter is unbounded.
While this is an interesting paper, with a fresh approach to estimate the complexity of natural language, there are some points, which should be improved or clarified.
Related work:
1. The objective used in this work appears to be equivalent to the β-VAE (β-VAE: Learning Basic Visual Concepts with a Constrained Variational Framework, Higgins et al. 2017) objective, with a normalizing flow prior, when replacing sequences $\overleftarrow{X}$ / $\overrightarrow{X}$-> with single data points X. While this work probes the regime where 0 < β < 1, β-VAE considers the regime of β >> 1. This analogy of β-VAE to the information bottleneck is e.g. expanded in ´Understanding disentangling in β-VAE´ by Burgess et al. 201. The authors should consider expanding on this analogy.
2. The idea of using past-future information bottleneck (PIB) as an objective for training neural networks for the task of sequence generation is not entirely new. There are a number of approaches for protein molecular dynamics trajectory generation by neural networks using an objective similar to the one in this work. E.g. ´Variational encoding of complex dynamics´ by Hernández et al. 2018, where their objective is equivalent to the one in this work, with M = 1 and lambda = 1, together with an additional autocorrelation loss term. This might be interesting to mention.
Unclear points:
3. After reading the paper, it is initially unclear, how future sequences X-> are sampled at different values of the parameter lambda. Reading the code seems to suggest, that a triple of neural networks is trained for each sample of lambda. Is this the case? If this is the case, this should constitute a large computational burden, when computing the rate-distortion curves, especially for natural language. How does this method compare to OCF in terms of training time / computation? The authors should clarify this in the manuscript.
4. In lines 289 - 293, the authors state they "create an LSTM network that estimates the marginal probability of future observation sequences", which they term $P_{\eta}$. The implementation of this is subtly different from what is stated in the text. In the code, the encoder LSTM $P_{\phi}$ doubles as this network. Its hidden state is used to compute Z, while its outputs are used to predict $\overrightarrow{X}$ directly. The fact that this is done subtly modifies the training objective to further include a further entropy term $log P_{eta}(\overrightarrow{X} | \overleftarrow{X})$, where $\eta$ and $\phi$ share weights. The authors should check, whether the proofs still hold in this case, and change the text to better reflect the actual implementation.
Proposed further experiments:
5. Throughout the paper, sequence length M is held constant (M = 15). This obviates the need to use a recurrent architecture like the LSTM used. LSTM cannot be efficiently parallelized, thus adding to the computational burden mentioned in point 3 above. Would it be conceivable to replace the LSTMs used by convolutional or fully connected neural networks, while retaining prediction accuracy, or otherwise reduce the computational burden? Furthermore, how does the length M influence the behaviour of the method? With the overall complexity of the presented method, the authors should provide some measure of ablation analysis.
6. This work uses a neural autoregressive flow prior (e.g. line 261). Experience has shown that for variational autoencoders simple priors often suffice to approximate a given distribution. Again, neural autoregressive flow priors add to the complexity of the method, and the authors should provide some measure of ablation analysis to justify its use.
7. As already stated in point 3 above, the way different values of lambda are sampled in this work should come with a high computational burden. This is unsatisfactory, for an otherwise very nice method. Would it be possible to condition both the encoder and decoder networks on lambda, and train at different lambda values, to condense the training to a single set of networks, for an a priori given set of lambdas? This is just a suggestion, which in my opinion would vastly improve the usability of the presented method. I would recommend the authors to try this.
8. It would be nice to have some qualitative examples of prediction results for the word-level natural language task at different rates, to be able to gain a better intuition of the effect of rate and predictivity on the quality of predictions.
Further comments:
9. The authors should also include information on the training process, e.g. early stopping, number of epochs, training time, and hardware in appendix A.10. It might be beneficial to align the notation for the approximate probability distributions with the standard in the VAE literature. The encoding distribution (variational posterior) is usually denoted by "q", here it is referred to as "P" (e.g. in line 198). Similarly, the prior is usually denoted by "p", here it is referred to as "q" (e.g. in line 210, or eq. 14).
11. In line 446 - 447, it is stated that "[n]eural encoders and decoders can use the continuous geometric structure of the code space to encode generalizations across different dimensions". While this is true, a rate of 15 nats (line 444) should also be easily feasible using discrete latent variables.
12. In line 274 - 275, it is stated, "optimization terminates once the value on the held-out set does not decrease any more". Please clarify, that the "held-out set" refers to the validation set.
Our overall recommendation would be to accept the manuscript for publication after ablation analyis has been presented and the rest of the comments have been address in the text.
Reviewer 2 Report
Review for "Estimating Predictive Rate-Distortion Curves via Neural Variational Inference"
*Summary: When trying to predict the future from observations made in the past, there is a trade-off between the accuracy of prediction and the amount of information about the past that is used (and stored) to predict. This trade-off is referred to as predictive rate-distortion. The authors introduce a novel method to estimate predictive rate-distortion by using neural networks in order to construct a summary code that encodes past observations and that can then be used to predict future observations, called neural predictive rate-distortion. The authors show convincingly that the outlined method surpasses the main prior method for estimating predictive rate-distortion from data which is called optimal causal filtering. They firstly show that unlike the latter, the new method is able to discover generalizations for sequences that were not been seen previously, which is highly important when it comes to data with a large state space such as natural language. Secondly, compared to the exponential memory demand of optimal causal filtering, the proposed method scales linearly with the length of the sequences observed in the past and thus sharply reducing the corresponding computational/storage costs. After formally introducing predictive rate-distortion, optimal causal filtering and neural predictive rate-distortion, the new method is validated for analytically known processes before first estimates for predictive rate-distortion of different natural languages are being presented.
*Recommendation: The paper is very well written, I have read it with great interest and pleasure (at times almost forgetting that is my job to review it). The relevant literature is adequately cited and discussed. Conclusions and interpretations are well supported by experimental results. While I am not a complete expert in this particular field, I was largely able to follow the formal description and the logic of the method and I have not been able to discover any errors/flaws whatsoever. All code for reproducing the results is available on Github which also deserves credit. I believe that this is an excellent paper that is highly appropriate for publication in Entropy. Therefore, I recommend to "accepting the paper in its present form" and would like my few comments below to be viewed as suggestions for improvement before publication. Kudos!
*Major points: none
*Minor points:
At times, the abbreviations disturb the reading flow of an already dense paper. Since Entropy does not have any length limitations, you could think about removing the abbreviations from the paper in order to improve the readability.
L73: "Causal" instead of "Causel"
L347: use the epsilon symbol instead of "epsilon symbol"
L367: do not abbreviate PCA
Figure 5: use the lambda symbol instead of "lambda"
L453: …"complexity of English sentences at the POS level" instead of "as the POS level"
Round 2
Reviewer 1 Report
I find the revised manuscript suitable for publication in Entropy.